# From Guessing to Placeholding: A Cost-Theoretic Framework for Uncertainty-Aware Code Completion

Liang Zhu[1]  Haolin Chen[1]  Lidong Zhao[1]  Xian Wu[1]

## Abstract

While Large Language Models (LLMs) have demonstrated exceptional proficiency in code completion, they typically adhere to a **Hard Completion (HC)** paradigm, compelling the generation of fully concrete code even amidst insufficient context. Our analysis of 3 million real-world interactions exposes the limitations of this strategy: 61% of the generated suggestions were either edited after acceptance or rejected despite exhibiting over 80% similarity to the user's subsequent code, suggesting that models frequently make erroneous predictions at specific token positions. Motivated by this observation, we propose **Adaptive Placeholder Completion (APC)**, a collaborative framework that extends HC by strategically outputting explicit placeholders at high-entropy positions, allowing users to fill directly via IDE navigation. Theoretically, we formulate code completion as a cost-minimization problem under uncertainty. Premised on the observation that filling placeholders incurs lower cost than correcting errors, we prove the existence of a critical entropy threshold above which APC achieves strictly lower expected cost than HC. We instantiate this framework by constructing training data from filtered real-world edit logs and design a cost-based reward function for reinforcement learning. Extensive evaluations across 1.5B–14B parameter models demonstrate that APC reduces expected editing costs from 19% to 50% while preserving standard HC performance. Our work provides both a theoretical foundation and a practical training framework for uncertainty-aware code completion, demonstrating that adaptive abstention can be learned end-to-end without sacrificing conventional completion quality.

[1]Tencent, Shenzhen, China. Correspondence to: Xian Wu <garethwu@tencent.com>.

*Proceedings of the 43rd International Conference on Machine Learning*, Seoul, South Korea. PMLR 306, 2026. Copyright 2026 by the author(s).

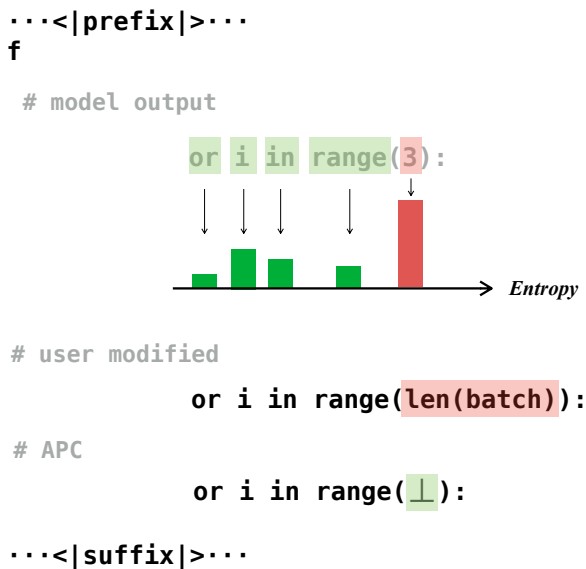

```
···<|prefix|>···
f

 # model output

   or i in range(3):

 # user modified

         or i in range(len(batch)):

 # APC

         or i in range(⊥):

···<|suffix|>···
```

*Figure 1.* Insufficient context leads to high-entropy predictions at specific positions (e.g., the loop boundary 3). Hard Completion forces a concrete guess, inducing costly hallucinations. Our proposed Adaptive Placeholder Completion (APC) instead generates reliable syntactic scaffolding while inserting a placeholder ($\bot$) at ambiguous positions, shifting the developer's burden from debugging hallucinations to seamlessly filling in blanks.

## 1. Introduction

Large Language Models (LLMs) have demonstrated exceptional capabilities (Achiam et al., 2023; Guo et al., 2025) in the field of code intelligence and revolutionized code-related applications (Li et al., 2022), excelling particularly in code generation (Guo et al., 2024; Zhu et al., 2024; Li et al., 2023b; Luo et al., 2023; Hui et al., 2024; Team et al., 2024; Roziere et al., 2023; Lozhkov et al., 2024) and the emerging paradigm of agentic coding (Yang et al., 2024; Wang et al., 2024; Jimenez et al., 2023). Nevertheless, among these applications, intelligent code completion (Raychev et al., 2014) represents a fundamentally more pervasive task. As a cornerstone feature embedded in development environments used by millions of programmers worldwide,

effective code completion directly translates to substantial gains in developer productivity, faster iteration cycles, and reduced development costs (Ziegler et al., 2022; Peng et al., 2023)—making it a critical area that demands continued research and optimization (Svyatkovskiy et al., 2020).

Despite its importance, current code completion practices remain constrained by a static operational paradigm—which we term **Hard Completion (HC)**: the model must invariably predict the complete, concrete implementation of a code segment. This "all-or-nothing" approach compels models to finalize every token, regardless of whether the context is sufficient or the intent is ambiguous.

Analysis of real-world developer interaction streams reveals that this forced speculation creates significant friction. Specifically, we observe a pervasive phenomenon where users reject a model's suggestion, or accept and immediately edit it, only to produce code that is semantically similar to the rejected inference. This typically occurs because the model, driven by the HC paradigm, attempts to "guess" specific details in regions where context is absent or ambiguity is high. Crucially, these guesses are not merely low-confidence predictions; models often exhibit miscalibrated confidence, hallucinating incorrect details with high certainty. When these "hard" predictions fail, the cognitive load required for a user to verify and surgically correct the mismatch often exceeds the cost of typing the code from scratch. In our analysis of over 3 million real-world interaction logs, we find that nearly 61% of cases involve such post-hoc editing or rejection-then-replication behaviors. While code completion aims to accelerate development, this paradigm paradoxically inflates the deletion and modification cost, thereby degrading the overall user experience. This observation begs a fundamental question:

*Can we transition code completion from a passive prediction task to an uncertainty-aware collaborative process?*

In this paper, we propose **Adaptive Placeholder Completion (APC)**, a novel paradigm that shifts the objective from solely maximizing token likelihood to minimizing the user's expected editing cost. In contrast to the fixed HC paradigm, APC empowers the model learning to adaptively determine whether to employ HC or **Placeholder Completion (PC)**— the latter inserting explicit placeholders (denoted as $\perp$) at positions of high uncertainty in the generated output. By generating syntactic skeletons that omit high-entropy details, APC strategically relinquishes completion authority to the user. This design, when coupled with IDE integration that supports seamless tab-navigation to placeholders, creates a highly efficient human-AI interaction loop. The model handles the predictable part, while the user fills in the ambiguous specifics, effectively reducing the friction of correcting wrong guesses.

Crucially, our work moves beyond heuristic engineering by establishing a rigorous theoretical framework for this paradigm. We formalize the human-AI code completion interaction as a **cost-minimization problem**. By decomposing the uncertainty in code generation into aleatoric uncertainty (inherent data noise) and epistemic uncertainty (model knowledge limitations), we derive the theoretical bounds of completion utility. We prove that in high-entropy regimes, there exists a definite entropy threshold above which APC yields a strictly lower expected cost than HC. This proof demonstrates that APC is not merely an alternative interface, but a mathematically superior strategy for minimizing user effort under uncertainty by trading the high cost of correcting specific errors for the significantly lower cost of filling a placeholder.

To materialize this framework, We construct a training dataset from real-world code completion user edit traces, carefully balancing the proportion of HC and PC instances. Additionally, we engage professional annotators to manually label PC examples, from which we construct a high-quality PC benchmark for evaluation. Our training pipeline begins with Supervised Fine-Tuning (SFT) to instill the placeholder generation capability, followed by Reinforcement Learning (RL) with a specifically designed cost-based reward function. This RL stage incentivizes the model to penalize costly hallucinations and optimize the precision of skeletal generation, thereby minimizing the theoretical expected cost.

Our contributions are summarized as follows:

- **Novel Paradigm:** We propose Adaptive Placeholder Completion (APC), a collaborative paradigm that dynamically switches between concrete prediction and placeholding, effectively mitigating the high cost of error correction in high-entropy regions.

- **Theoretical Framework:** We model code completion as a cost-minimization problem, proving that APC is theoretically superior to HC in high-entropy scenarios driven by aleatoric or epistemic uncertainty. Grounded in this framework, we design a tailored cost-based reward function for reinforcement learning that directly optimizes the expected user cost.

- **Extensive Empirical Analysis:** We establish a complete pipeline encompassing training data construction, theory-aligned training methodology, and rigorous evaluation benchmarks with tailored metrics. Through extensive experiments across 4 distinct code LLM families spanning 1.5B to 15B parameters, we demonstrate both the learnability and effectiveness of APC, showing significant improvements in completion efficiency and user experience compared to standard baselines.

## 2. Related Work

### 2.1. Code Completion Paradigms

The evolution of automated code completion has progressed from early statistical modeling to context-aware LLMs. Raychev et al. (2014) pioneered the use of statistical n-gram models to predict code sequences, demonstrating that programming languages exhibit high repeatability similar to natural languages. The advent of Transformer-based architectures revolutionized this domain, with Chen et al. (2021) introducing Codex and shifting the evaluation paradigm from surface-level matching to functional correctness. While standard causal models generate strictly in a **Left-to-Right** paradigm, subsequent innovations addressed the need for infilling and bidirectional context: Bavarian et al. (2022) formalized the **Fill-in-the-Middle (FIM)** objective with the `[Prefix, Suffix, Middle]` formulation, and concurrently, Fried et al. (2022) proposed InCoder using causal masking with sentinel tokens, collectively enabling models to fill missing code segments within arbitrary contexts. To further mitigate uncertainty arising from cross-file dependencies, Zhang et al. (2023) introduced RepoCoder, which employs iterative retrieval-augmented generation to leverage repository-level context. Despite these advancements in generation capability and context utilization, all aforementioned approaches still adhere to the HC paradigm. Strategies for adaptively managing high-entropy scenarios through explicit placeholder mechanisms remain largely underexplored.

### 2.2. Uncertainty Estimation and Selective Generation

Machine learning uncertainty is formally decomposed into **aleatoric** and **epistemic** types (Hüllermeier & Waegeman, 2021), with recent work quantifying this via **semantic uncertainty** (Kuhn et al., 2023) to overlook trivial surface variations. Strategies to leverage such uncertainty typically fall into two categories. **Abstention** methods (Tomani et al., 2024; Shi et al., 2024) employ binary decision policies——withholding predictions entirely or suppressing hallucinations via context-aware decoding when confidence is low. While maximizing precision, complete silence disrupts developer workflow in real-time completion tasks. Conversely, **skeleton-based** approaches (Zheng et al., 2023; Li et al., 2023a; Zan et al., 2022) separate structure from detail through multi-stage pipelines (outline-then-fill, retrieve-then-edit) or continual pre-training on code sketches. However, these methods treat skeletons as intermediate representations for eventual full-code generation, hidden from end-users, and re-infilling mechanisms (Wei et al., 2023) operate only as post-hoc repair. Our APC paradigm differs fundamentally by enabling the model to **implicitly learn** when to perform hard completion versus placeholder insertion through end-to-end training, adaptively internaliz-

ing the cost-minimization decision boundary without explicit entropy computation. Unlike multi-stage pipelines, APC operates in a **single inference pass**, with IDE integration recognizing the placeholder token to enable seamless tab-navigation, creating a direct human-AI interaction loop. We prove the existence of a critical entropy threshold above which placeholders provably reduce expected editing cost, providing the first rigorous justification for uncertainty-aware code completion that preserves workflow while inviting human collaboration.

### 2.3. Reinforcement Learning for LLMs

While supervised fine-tuning (SFT) instills basic capabilities, aligning models with human intent typically requires Reinforcement Learning from Human Feedback (RLHF) (Ouyang et al., 2022). In the software engineering domain, this alignment often leverages the executable nature of code. CodeRL (Le et al., 2022) and PPOCoder (Shojaee et al., 2023) integrate compiler feedback and unit test results directly into the reward signal, optimizing policies to maximize functional correctness. To mitigate the computational instability and memory overhead of standard PPO (Schulman et al., 2017), recent research has shifted towards direct optimization methods like DPO (Rafailov et al., 2023). Notably, DeepSeekMath (Shao et al., 2024) introduced **Group Relative Policy Optimization (GRPO)**, which eliminates the need for a separate value function critic by normalizing rewards within a sampled group of outputs. This approach has proven exceptionally effective for reasoning-heavy tasks where outcome verification (like passing a test case) is binary and objective. However, existing alignment strategies for code focus predominantly on correctness or style. They implicitly assume a "one-shot" success scenario. In contrast, our work adapts GRPO to a collaborative setting. Instead of solely maximizing test pass rates, we design a novel reward function grounded in our theoretical **expected editing cost**. This aligns the model not just with the code's syntax, but with the user's editing behavior, incentivizing the model to use placeholders precisely when the "cost of guessing" outweighs the "cost of typing."

## 3. Theoretical Framework

In this section, we formalize the code completion process as a cost-minimization problem, showing that under specific entropy conditions, the proposed Adaptive Placeholder Completion (APC) paradigm yields a strictly lower expected user cost compared to traditional Hard Completion (HC). Notations used in this section can be found in Appendix A.

### 3.1. Problem Formulation

Given a code context $c$, the objective is to generate a sequence $\mathbf{y}^* = (y_1^*, \ldots, y_n^*)$. The code completion task can

be viewed as a decision-making problem under information constraints. The uncertainty encountered by the model stems primarily from two distinct sources: **Epistemic** and **Aleatoric** uncertainty. The former arises from reducible limitations in model capacity or knowledge—specifically, the inability to infer the correct solution from available context. This can be mitigated by stronger architectures or broader training data (e.g., correct API usage known to a model trained on relevant libraries but unknown to a weaker one). Conversely, inherent ambiguity within the context where the information is insufficient to determine a unique ground truth.

We define two distinct completion strategies for the $i$-th token $\hat{y}_i$:

- **Hard Completion (HC):** The model predicts a concrete token $\hat{y}_i \in \mathcal{V}$.

- **Placeholder Completion (PC):** The model generates a placeholder token, denoted as $\perp$.

We introduce a cost function $C(\hat{\mathbf{y}}, \mathbf{y}^*)$ to quantify the user's effort to get the ideal code sequence $\mathbf{y}^*$ during a trigger of completion $\hat{\mathbf{y}}$. So the cost at step $i$ is defined as:

$$C_i(\hat{y}_i, y_i^*) = \begin{cases} 0, & \text{if } \hat{y}_i = y_i^* \\ C_{\text{HC}}, & \text{if } \hat{y}_i \neq y_i^* \wedge \hat{y}_i \in \mathcal{V} \\ C_{\text{PC}}, & \text{if } \hat{y}_i = \perp \end{cases} \quad (1)$$

At token position $i$, an **Exact Match (EM)** between the model's prediction and the ground truth yields zero cost. Conversely, in the event of a discrepancy, the incurred cost bifurcates based on the adopted strategy: Hard Completion entails a penalty of $C_{\text{HC}}$, whereas Adaptive Placeholder Completion incurs a fixed cost of $C_{\text{PC}}$. Considering that the aggregate cost of an HC error comprises error identification, deletion, and cursor navigation, whereas the cost of a placeholder is restricted solely to manual infilling; and premised on the observation that models achieve high EM rates in low-entropy regions, we propose the following assumption:

**Assumption 3.1.** The interaction cost of filling an explicit placeholder is strictly less than the aggregate cost (comprising error identification, deletion, and navigation) of correcting a hallucinated error, i.e.,

$$0 < C_{\text{PC}} < C_{\text{HC}} \quad (2)$$

*Remark* 3.2. We acknowledge that for minor deviations (e.g., typos), editing a prediction might be faster than typing from scratch. However, our framework specifically targets **high-entropy regimes**. In these contexts, errors typically manifest as hallucinations or logic mismatches rather than near-misses. The cognitive load required to verify and reject such misleading suggestions (the "anchoring effect") further validates the inequality $C_{\text{PC}} < C_{\text{HC}}$.

### 3.2. Expected Cost Analysis

Since the ground truth $\mathbf{y}^*$ remains latent during the inference phase, the optimal completion strategy cannot be determined deterministically. Consequently, we treat the user's intended code as a stochastic variable and the task can be regarded as an **expected cost minimization** problem. Our objective is to optimize the completion strategy by evaluating the cost function in expectation over all plausible realizations of the ground truth, governed by the intrinsic conditional probability distribution $P(\cdot \mid c)$:

$$\mathcal{J}(\hat{\mathbf{y}}) = \mathbb{E}_{\mathbf{y}^* \sim P(\cdot \mid c)} [C(\hat{\mathbf{y}}, \mathbf{y}^*)] \quad (3)$$

For HC, to facilitate analytical tractability, we assume the model employs greedy decoding at each step, selecting the token with maximal probability:

$$\hat{y}_i^{\text{HC}} = \arg\max_{v \in \mathcal{V}} P(\hat{y}_i = v \mid c, \hat{y}_{<i}),$$
$$\text{where} \quad P_{\max}^{(i)} = \max_{v \in \mathcal{V}} P(\hat{y}_i = v \mid c, \hat{y}_{<i}) \quad (4)$$

Under this greedy assumption, the expected cost of HC can be expressed from Appendix B.1.1:

$$\mathcal{J}^{\text{HC}}(\hat{\mathbf{y}}) = \mathbb{E}^{\text{HC}}[C(\hat{\mathbf{y}}, \mathbf{y}^*)]$$
$$= \sum_i C_{\text{HC}} \cdot (1 - P_{\max}^{(i)}) \quad (5)$$

For APC, we first introduce a binary mask vector $\mathbf{m}$ to indicate whether the model outputs a placeholder at position $i$ under the strategy:

$$m_i = \begin{cases} 1, & \text{if } \hat{y}_i = \perp \\ 0, & \text{if } \hat{y}_i \neq \perp \end{cases} \quad (6)$$

Subsequently, the expected cost of APC can be expressed from Appendix B.1.2:

$$\mathcal{J}^{\text{APC}}(\hat{\mathbf{y}}) = \mathbb{E}^{\text{APC}}[C(\hat{\mathbf{y}}, \mathbf{y}^*)]$$
$$= \sum_i \left[ m_i \cdot C_{\text{PC}} + (1 - m_i) \cdot C_{\text{HC}} \cdot (1 - P_{\max}^{(i)}) \right] \quad (7)$$

### 3.3. Optimality Condition for APC

Having established the expected cost formulations for both strategies, we now derive the condition under which PC yields a strictly lower expected cost than HC. We define the *expected benefit* of adopting APC over HC as:

$$\Delta \mathcal{J} = \mathcal{J}^{\text{HC}}(\hat{\mathbf{y}}) - \mathcal{J}^{\text{APC}}(\hat{\mathbf{y}}) \quad (8)$$

Substituting Equation (5) and Equation (7), we obtain:

$$\Delta \mathcal{J} = \sum_i m_i \cdot \left[ C_{\text{HC}} \cdot (1 - P_{\max}^{(i)}) - C_{\text{PC}} \right] \quad (9)$$

Let us define the *per-position benefit* as:

$$\delta_i = C_{\text{HC}} \cdot (1 - P_{\max}^{(i)}) - C_{\text{PC}} \quad (10)$$

which is detail in Appendix B.1.3. Equation (9) reveals that the overall benefit $\Delta \mathcal{J}$ is a weighted sum of per-position benefits, where the weights are determined by the mask vector $\mathbf{m}$. Crucially, $\Delta \mathcal{J} > 0$ (i.e., APC is superior) if and only if there exists at least one position $i$ where both conditions hold: (1) the model chooses to insert a placeholder ($m_i = 1$), and (2) the per-position benefit is positive ($\delta_i > 0$). This leads to the following optimality condition:

$$\delta_i > 0 \iff C_{\text{HC}} \cdot (1 - P_{\max}^{(i)}) > C_{\text{PC}}$$
$$\iff P_{\max}^{(i)} < 1 - \frac{C_{\text{PC}}}{C_{\text{HC}}} \quad (11)$$

In other words, at position $i$, the placeholder strategy is optimal precisely when the model's confidence $P_{\max}^{(i)}$ falls below the critical threshold $1 - \frac{C_{\text{PC}}}{C_{\text{HC}}}$, which is determined solely by the cost ratio. This condition establishes a clear decision boundary: placeholders should be used when uncertainty is sufficiently high. While Equation (11) provides a probability-based criterion, in practice, directly measuring $P_{\max}^{(i)}$ requires access to the model's internal distribution. To establish a more interpretable and actionable criterion, we now bridge the gap between model confidence and Shannon entropy. Intuitively, high entropy at position $i$ implies a flat probability distribution over the vocabulary, leading to low $P_{\max}^{(i)}$. We formalize this relationship and prove the existence of a critical entropy threshold above which APC is guaranteed to outperform HC.

Suppose there exists a token position $i$ where the entropy is sufficiently high, with $K$ tokens as candidatesa and share the probability mass almost equally. In this situation, we can get:

$$P_{\max}^{(i)} \approx \frac{1}{K} = e^{-H} \quad (12)$$

Set the threshold:

$$\theta = 1 - \frac{C_{\text{PC}}}{C_{\text{HC}}} \quad (13)$$

The proof target can be transformed into:

$$\exists H_i : e^{-H_i} - \theta < 0 \quad (14)$$

We denote:

$$f(H) = e^{-H} - \theta \quad (15)$$

Then we can get the main theoretical result of this paper:

**Theorem 3.3.** *Given Assumption 3.1, there exists a unique critical entropy threshold:*

$$H^* = \ln \left( \frac{C_{\text{HC}}}{C_{\text{HC}} - C_{\text{PC}}} \right) \quad (16)$$

*such that for any token position $i$ in the completion sequence where the conditional entropy $H(y_i^* \mid c, y_{<i}^*)$ exceeds $H^*$, the Adaptive Placeholder Completion strategy achieves strictly lower expected cost than Hard Completion:*

$$H(y_i^* \mid c, y_{<i}^*) > H^*$$
$$\implies \mathbb{E}[C_i(\hat{y}_i^{APC}, y_i^*)] < \mathbb{E}[C_i(\hat{y}_i^{HC}, y_i^*)] \quad (17)$$

**Corollary 3.4.** *If there exists at least one token position $i$ in the completion sequence where the conditional entropy satisfies $H(y_i^* \mid c, y_{<i}^*) > H^*$, then the Adaptive Placeholder Completion strategy achieves strictly lower overall expected cost than Hard Completion:*

$$\mathcal{J}^{APC}(\hat{\mathbf{y}}) < \mathcal{J}^{HC}(\hat{\mathbf{y}}) \quad (18)$$

**From Theory to Practice** While our derivation establishes an explicit threshold $H^*$, it is crucial to recognize that this formula relies on a maximum entropy (uniform distribution) assumption to provide a worst-case analytical bound. In reality, real-world LLM token distributions are highly non-uniform and long-tailed. Furthermore, the local cost ratio $C_{\text{PC}}/C_{\text{HC}}$ dynamically shifts across varying coding contexts. Consequently, applying a static mathematical threshold is fundamentally intractable for real-world deployment. Rather than viewing this as a limitation, we treat this analytical proof as the theoretical motivation for our methodology. Because the optimal decision boundary is a dynamic, high-dimensional surface, we formulate the cost-minimization objective as an inductive bias and leverage Supervised Fine-Tuning (SFT) combined with Reinforcement Learning (RL). This end-to-end learning approach forces the model to **implicitly** internalize the complex decision surface without relying on flawed heuristic approximations.

# 4. Experiments

## 4.1. Experimental Setup

### 4.1.1. DATASETS

**Training** We construct our training dataset from real-world user interaction logs, where each sample pairs a model prediction $\hat{\mathbf{y}}$ with the user-edited ground truth $\mathbf{y}$. Samples with EM between $\hat{\mathbf{y}}$ and $\mathbf{y}$ serve as HC training instances. For the remaining samples, we align $\hat{\mathbf{y}}$ and $\mathbf{y}$ token-wise, replacing mismatched positions with the placeholder to construct APC training instances. This yields 5,000 HC samples and 10,000 PC samples, ensuring balanced exposure to both completion paradigms while emphasizing the model's ability to strategically abstain under uncertainty.

**Evaluation** For HC, we adopt the established HumanEval Infilling Benchmark (Bavarian et al., 2022), comprising 1,033 single-line and 5,815 multi-line completion instances.

*Table 1.* APC training results across model families and scales. SFT introduces PC capabilities, with GRPO further enhancing performance. All models maintain 100% HCR and stable HC test performance while consistently improving Precision, ES, F1, and Cost metrics.

| MODEL | | | HC TEST | | | PC TEST | | | | | |
|---|---|---|---|---|---|---|---|---|---|---|---|
| | | | HCR | EM | ES | PCR | PREC | EM | ES | F1 | COST↓ |
| QWEN2.5-CODER | 1.5B | BASE | 100 | 41.88 | **77.36** | 0.19 | 0.00 | 0.00 | 21.76 | 0.00 | 72.93 |
| | | SFT | 100 | **44.55** | 74.96 | 35.37 | 36.76 | 13.00 | 70.32 | 19.21 | 38.08 |
| | | GRPO | 100 | 41.02 | 70.90 | **60.23** | **45.71** | **27.53** | **74.95** | **34.37** | **36.75** |
| | 3B | BASE | 100 | 46.74 | **83.19** | 2.49 | 0.00 | 0.00 | 54.16 | 0.00 | 45.42 |
| | | SFT | 100 | **48.57** | 82.73 | 12.81 | 41.79 | 5.35 | 67.84 | 9.49 | 40.61 |
| | | GRPO | 100 | 46.49 | 77.97 | **43.59** | **50.88** | **22.18** | **76.56** | **30.89** | **36.60** |
| | 7B | BASE | 100 | 24.29 | 54.68 | 0.19 | 0.00 | 0.00 | 35.10 | 0.00 | 60.75 |
| | | SFT | 100 | 49.24 | 83.27 | 63.10 | 45.76 | 28.87 | 76.24 | 35.40 | 37.73 |
| | | GRPO | 100 | **51.08** | **86.07** | **68.26** | **47.06** | **32.12** | **78.72** | **38.18** | **36.38** |
| | 14B | BASE | 100 | 30.35 | 57.77 | 0.00 | 0.00 | 0.00 | 31.72 | 0.00 | 62.82 |
| | | SFT | 100 | **51.32** | **84.21** | 53.35 | 48.39 | 25.81 | 76.53 | 33.67 | 35.15 |
| | | GRPO | 100 | 47.04 | 79.75 | **65.20** | **50.15** | **32.70** | **79.05** | **39.58** | **34.54** |
| CODEGEMMA | 2B | BASE | 100 | **43.24** | **80.56** | 0.00 | 0.00 | 0.00 | 44.25 | 0.00 | 55.85 |
| | | SFT | 100 | 32.67 | 40.31 | 13.38 | 30.00 | 4.02 | 53.44 | 7.08 | 44.06 |
| | | GRPO | 100 | 43.10 | 74.81 | **51.63** | **37.41** | **19.31** | **72.11** | **25.47** | **37.06** |
| | 7B | BASE | 100 | **40.51** | **77.70** | 0.00 | 0.00 | 0.00 | 44.72 | 0.00 | 48.92 |
| | | SFT | 100 | 34.21 | 66.94 | 38.24 | 31.00 | 11.85 | 65.13 | 17.15 | 38.14 |
| | | GRPO | 100 | 40.10 | 74.98 | **47.61** | **42.57** | **20.27** | **74.57** | **27.46** | **34.71** |
| CODELLAMA | 7B | BASE | 100 | 11.54 | 42.78 | 0.00 | 0.00 | 0.00 | 24.75 | 0.00 | 68.14 |
| | | SFT | 100 | **37.10** | 70.14 | 58.32 | 32.46 | 18.93 | 71.00 | 23.91 | 38.46 |
| | | GRPO | 100 | 36.83 | **70.48** | **61.57** | **34.16** | **21.03** | **74.13** | **26.04** | **37.59** |
| | 13B | BASE | 100 | 17.07 | 48.99 | 0.00 | 0.00 | 0.00 | 26.98 | 0.00 | 65.99 |
| | | SFT | 100 | 31.18 | 63.76 | 58.32 | 31.48 | 18.36 | 71.63 | 23.19 | 37.29 |
| | | GRPO | 100 | **35.72** | **68.49** | **64.05** | **36.12** | **23.14** | **75.53** | **28.21** | **36.68** |
| STARCODER2 | 3B | BASE | 100 | 38.46 | 75.47 | 0.00 | 0.00 | 0.00 | 26.62 | 0.00 | 67.36 |
| | | SFT | 100 | 42.20 | 77.15 | 49.52 | 33.65 | 16.67 | 66.22 | 22.29 | 39.02 |
| | | GRPO | 100 | **43.40** | **79.23** | **59.05** | **37.90** | **22.38** | **69.26** | **28.14** | **37.57** |
| | 7B | BASE | 100 | 42.90 | 79.78 | 0.00 | 0.00 | 0.00 | 30.75 | 0.00 | 62.81 |
| | | SFT | 100 | **45.67** | **81.65** | 38.10 | 33.75 | 12.86 | 68.36 | 18.62 | 37.31 |
| | | GRPO | 100 | 38.98 | 74.16 | **49.33** | 11.47 | **20.46** | **75.03** | **27.40** | **35.40** |
| | 15B | BASE | 100 | 40.53 | 78.43 | 0.00 | 0.00 | 0.00 | 22.40 | 0.00 | 69.89 |
| | | SFT | 100 | 44.04 | **79.51** | 65.71 | 44.93 | 29.52 | 75.08 | 35.63 | 38.30 |
| | | GRPO | 100 | **44.05** | 78.69 | **67.88** | **45.35** | **30.78** | **77.49** | **36.67** | **35.82** |

For APC, we follow the same data construction methodology as the training set, with an additional quality control step: professional annotators manually verify each placeholder to ensure its necessity, filtering out cases where concrete predictions would be unambiguous. This process yields 523 curated test samples with validated placeholders.

We performed reciprocal cross-annotation among annotators on a subset of the PC benchmark. The inter-annotator agreement using **Fleiss' Kappa**, which resulted in a score of **0.7994**. According to the standard Landis & Koch scale, this score represents the highest end of "**Substantial Agreement**" and borders on "Almost Perfect" (0.81). This confirms that the benchmark construction reflects consistent human judgment and reliably captures real-world uncertainty scenarios.

### 4.1.2. BASELINES

**Models** We evaluate our approach across four widely-adopted code model families: Qwen2.5-Coder (1.5B[1], 3B[2], 7B[3], 14B[4]) (Hui et al., 2024), CodeGemma (2B[5], 7B[6]) (Team et al., 2024), CodeLlama (7B[7], 13B[8]) (Roziere

---

[1]https://huggingface.co/Qwen/Qwen2.5-Coder-1.5B-Instruct
[2]https://huggingface.co/Qwen/Qwen2.5-Coder-3B-Instruct
[3]https://huggingface.co/Qwen/Qwen2.5-Coder-7B-Instruct
[4]https://huggingface.co/Qwen/Qwen2.5-Coder-14B-Instruct
[5]https://huggingface.co/google/codegemma-2b
[6]https://huggingface.co/google/codegemma-7b
[7]https://huggingface.co/meta-llama/CodeLlama-7b-hf
[8]https://huggingface.co/meta-llama/CodeLlama-13b-hf

*Table 2.* APC reduces editing actions and time consumption by over 30% while producing slightly longer completions, indicating that the efficiency gain stems from better placeholder placement rather than shorter outputs. All values are averages per completion instance.

| Metric | HC-only | APC | $\Delta$ |
|---|---|---|---|
| **Cursor Moves** (Keystrokes + Edit) | 4.89 | **3.06** | $\downarrow$ 37.4% |
| **Time for Read & Edit** (ms) | 3280.49 | **2255.17** | $\downarrow$ 31.2% |
| **Output Length** (Tokens) | 11.05 | **12.53** | + 1.48 |
| **Output Length** (Lines) | 1.34 | **1.52** | + 0.18 |

et al., 2023), and StarCoder2 (3B[9], 7B[10], 15B[11]) (Lozhkov et al., 2024), spanning model sizes from 1.5B to 15B parameters. This diverse selection enables comprehensive assessment of APC's effectiveness across different architectures and model scales.

**Methods** We compare three model variants: **(1) Base**: the pretrained foundation model; **(2) SFT**: the base model fine-tuned with supervised learning on our curated dataset; **(3) GRPO**: the SFT model further optimized via RL training using a cost-based reward function detailed in Appendix C. Both SFT and GRPO training stages utilize identical data sources and mixture ratios to isolate the effect of our designed reward function. All experiments are conducted on two machines, each equipped with 8 NVIDIA H20 GPUs.

### 4.1.3. EVALUATION METRICS

We evaluate model performance using the following metrics:

**Hard Completion Rate (HCR)**: The proportion of model outputs containing no placeholders, indicating the frequency of concrete predictions, denoted as .

**Placeholder Completion Rate (PCR)**: The proportion of model outputs containing at least one placeholder, measuring abstention frequency. By definition, HCR + PCR = 1 for any test set.

**Exact Match (EM)**: The percentage of model predictions that exactly match the ground truth. On placeholder-annotated test sets, this is equivalent to Recall.

**Edit Similarity (ES)**: A character-level similarity metric between model predictions and ground truth based on normalized Levenshtein distance, where placeholder tokens count as atomic character units.

**Precision**: Among predictions containing placeholders, the percentage that achieve exact match with the ground truth, assessing the appropriateness of placeholder insertion, denoted as Pred.

**F1 Score**: The harmonic mean of EM (Recall) and Precision,

providing a balanced measure of placeholder prediction quality, denoted as F1.

**Expected Cost**: The normalized editing cost required to transform model predictions into the real user edit nswer, defined as $1 - \mathrm{ES}(\hat{y}, y_{\mathrm{user}}^*)$. Lower values indicate reduced user effort, denoted as Cost.

### 4.2. Main Results

As illustrated in Table 1, models demonstrate clear progression in learning APC capabilities. Before training, placeholder generation is virtually absent across all model families and scales——a natural consequence of their standard pre-training on holistic completion. SFT introduces PC capabilities, with all models showing improvements in Precision, ES, F1, and Cost. GRPO training further strengthens these gains with consistent metric improvements.

Two observations are particularly noteworthy. First, performance on HC test remains stable or even improves throughout training, without obvious degradation in standard completion quality. Second, and more significantly, all models maintain perfect HCR (100%) across all training stages. While PCR values on PC test remain moderate rather than extreme, this pattern reflects ideal APC behavior: models preserve robust hard completion capabilities while judiciously inserting placeholders in appropriate contexts. This balanced performance validates that models have successfully learned to adaptively select between HC and PC strategies based on contextual signals, rather than developing a systematic bias toward either paradigm.

### 4.3. Real-world Interaction Cost Validation

To provide objective, physical evidence of developer effort reduction, we concluded a large-scale online A/B test analyzing **1.8 million real user interactions** (1.2M from the HC-only model and 0.6M from our APC model). The objective measurements are presented in Table 2. The APC paradigm **reduces cognitive validation time and physical typing friction by 31–37%**, while simultaneously increasing the average output of tokens and lines. This strongly supports our core claim: that guiding users with a correct skeleton and targeted placeholders requires less real-world effort than inspecting, rejecting, and surgically modifying a

---

[9]https://huggingface.co/bigcode/starcoder2-3b
[10]https://huggingface.co/bigcode/starcoder2-7b
[11]https://huggingface.co/bigcode/starcoder2-15b

*Table 3.* Ablation results for placeholder training data and cost-based reward design on Qwen2.5-Coder-7B-Instruct.

| METHOD | BASE | HC TEST | | | PC TEST | | | | | |
|---|---|---|---|---|---|---|---|---|---|---|
| | | HCR | EM | ES | PCR | PREC | EM | ES | F1 | COST |
| INSTRUCT | - | 100 | 24.29 | 54.68 | 0.19 | 0.00 | 0.00 | 35.10 | 0.00 | 60.75 |
| $\text{SFT}_{\text{USER}}$ | INSTRUCT | 100 | 47.47 | 79.50 | 0.00 | 0.00 | 0.00 | 64.54 | 0.00 | 39.29 |
| $\text{SFT}_{\text{APC}}$ | INSTRUCT | 100 | 49.24 | 83.27 | 63.10 | 45.76 | 28.87 | 76.24 | 35.40 | 37.73 |
| $\text{GRPO}_{\text{ES}}$ | $\text{SFT}_{\text{APC}}$ | 100 | 50.61 | 85.18 | 67.11 | 45.87 | 30.78 | 77.67 | 36.84 | 37.02 |
| $\text{GRPO}_{\text{APC}}$ | $\text{SFT}_{\text{APC}}$ | 100 | 51.08 | 86.07 | 68.26 | 47.06 | 32.12 | 78.72 | 38.18 | 36.38 |

hallucinated prediction.

# 5. Ablation and Analysis

## 5.1. Superiority of Placeholder Data

To further validate the effectiveness of the APC strategy, we conduct a controlled comparison using Qwen2.5-Coder-7B-Instruct as the base model. We contrast two SFT approaches: (1) **SFT-User**, which directly fine-tunes on real user edit traces as ground truth, and (2) **SFT-APC**, which trains on our curated dataset with explicit placeholders. As shown in Table 3, SFT-APC consistently outperforms SFT-User across all metrics. Notably, SFT-APC maintains superior performance on both PC test and HC test, demonstrating that placeholder-augmented training data not only enables effective placeholder generation but also preserves—and even enhances——hard completion quality. These results confirm that explicitly incorporating placeholder examples in training data is essential for learning the APC strategy effectively.

## 5.2. Effectiveness of Cost-Based Reward

To empirically assess the contribution of our cost-based reward design, we perform controlled RL experiments using SFT-APC as the initialization. We evaluate two reward formulations: (1) **GRPO-ES**, which adopts ES between generated completions and ground truth, and (2) **GRPO-APC**, which leverages our theoretically grounded cost-based reward function targeting expected user edit cost.

Results presented in Table 3 reveal that both RL variants achieve gains over the supervised baseline, indicating that policy refinement through RL is beneficial. Critically, however, GRPO-APC demonstrates comprehensive superiority over GRPO-ES across all evaluation metrics—higher Precision, ES, and F1 scores alongside reduced Cost. This substantiates that our cost-based reward function, which explicitly models user interaction efficiency rather than mere output similarity, more effectively guides the model toward adaptive placeholder completion strategy. The results validate that aligning the training objective directly with cost minimization is essential for optimal APC performance.

## 5.3. Why Implicit Learning?

As discussed at the end of Section 3, $H^*$ is not a fixed constant but a dynamic, context-dependent value. This motivates training models to implicitly learn the decision boundary via SFT and RL. To validate this premise, we design a controlled experiment applying fixed thresholds to post-process base model outputs. Using the Qwen2.5-Coder-Instruct family without any APC training, we compute the conditional entropy and confidence at each token position during inference on the PC benchmark. We then sweep various entropy and confidence thresholds—replacing tokens exceeding the entropy threshold or falling below the confidence threshold with placeholders—and evaluate the modified outputs using PCR and Cost metrics. The resulting performance curves are shown in Figure 2, with horizontal dashed lines marking each base model's original Cost.

As the entropy threshold increases (or confidence threshold decreases), PCR drops monotonically—fewer positions trigger placeholder insertion. Cost initially decreases as effective placeholders take effect, yet critically, the Cost curves for all model scales converge to a value nearly identical to the base model's original inference cost regardless of threshold configuration. This demonstrates that naively applying a fixed threshold yields no net benefit: misplaced placeholders offset the gains from well-placed ones. In contrast, the SFT and GRPO variants in Table 1 achieve substantially lower Cost, empirically validating that the optimal entropy threshold is inherently context-dependent and cannot be captured by a universal constant—only through implicit learning can models internalize the dynamic decision boundary required for true cost minimization.

## 5.4. Entropy of PC Benchmark

Our theoretical framework rests on a foundational assumption: regions requiring placeholders inherently exhibit elevated information entropy. To empirically substantiate this premise, we conduct a systematic entropy analysis of our PC benchmark using Qwen2.5-Coder-7B-Instruct as the reference model. For each test sample, we separately compute the average conditional entropy over tokens in placeholder-designated regions and in hard completion regions. Impor-

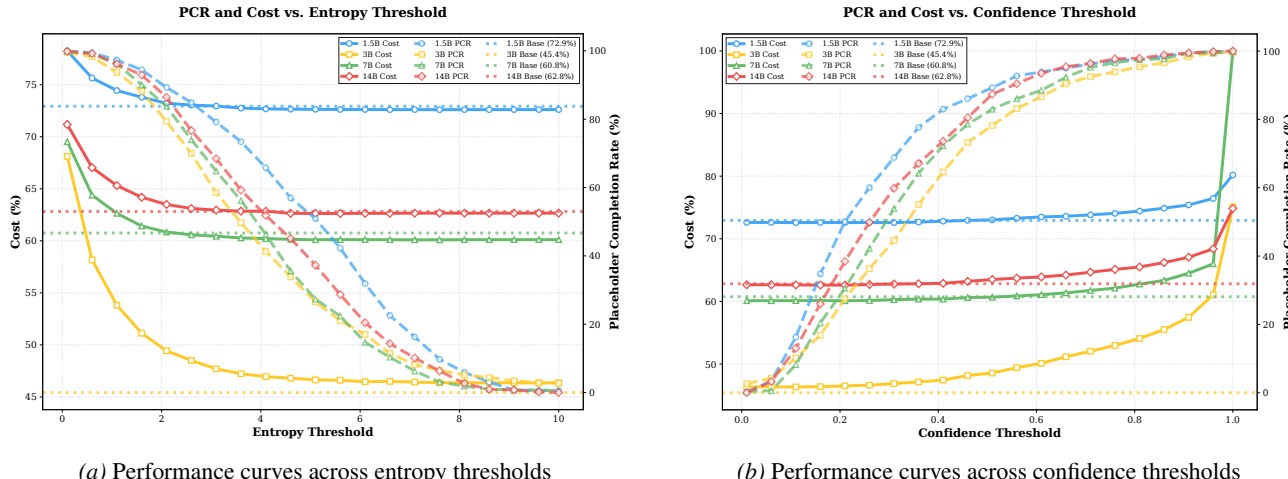

*(a)* Performance curves across entropy thresholds

*(b)* Performance curves across confidence thresholds

*Figure 2.* Fixed-threshold post-processing on base Qwen2.5-Coder-Instruct models. (a) Entropy threshold sweeps and (b) confidence threshold sweeps both show that Cost converges to the base model level (dashed line) regardless of threshold values, demonstrating that explicit thresholds cannot effectively reduce cost without implicit learning.

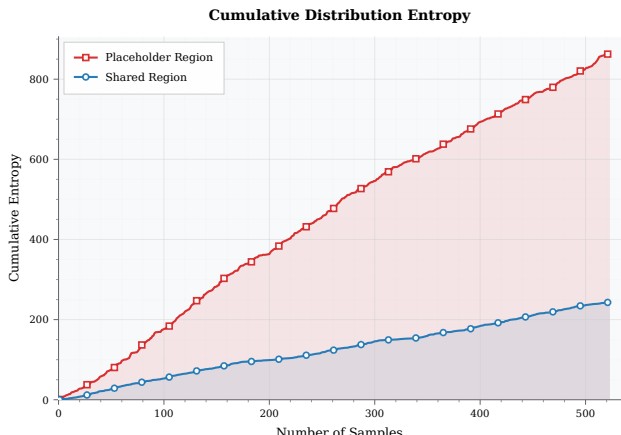

*Figure 3.* Cumulative average entropy for PC regions vs. HC regions. PC regions show substantially higher entropy accumulation, confirming that human-annotated placeholders align with high-uncertainty positions.

tantly, these entropy values are derived via autoregressive computation on the original ground truth sequence prior to placeholder substitution, ensuring that the model's output distribution accurately captures the positional uncertainty inherent in each token. We then track the cumulative average entropy as dataset size increases, contrasting the accumulation patterns between placeholder and hard completion regions.

As shown in Figure 3, the cumulative curve for PC regions exhibits a substantially steeper slope compared to

HC regions, indicating a systematic shift toward higher entropy values. This pronounced difference demonstrates that placeholder-annotated positions universally correspond to high-uncertainty contexts—human annotators consistently identify high-entropy scenarios for placeholder insertion, precisely aligning with our theoretical assumption. This empirical observation provides direct validation of our cost-theoretic framework: the regions where APC employs placeholders are indeed the high-entropy regimes where Theorem 3.3 guarantees cost optimality.

## 6. Conclusion

We introduced Adaptive Placeholder Completion (APC), a collaborative framework that transforms code completion into an uncertainty-aware decision-making process. By formalizing completion as a cost minimization problem, we derived an analytical entropy threshold, which serves as the theoretical motivation for our end-to-end optimization. Our training methodology combines supervised fine-tuning with cost-based reinforcement learning, enabling models to implicitly internalize dynamic decision boundaries. Experiments across four model families demonstrate that APC preserves HC ability while significantly reducing proxy editing costs. Crucially, a large-scale A/B test involving 1.8 million real user interactions validates our paradigm's practical impact: APC reduces physical editing actions and time consumption by 31%–37% without sacrificing output length—even generating slightly more comprehensive syntactic scaffolding. This work establishes a pragmatic and principled foundation for integrating uncertainty quantification into real-world code intelligence systems.

## Impact Statement

By shifting code LLMs toward uncertainty-aware collaboration, our Adaptive Placeholder Completion (APC) paradigm reduces cognitive friction and mitigates the societal risk of developers unknowingly integrating subtle, AI-generated vulnerabilities into production codebases.

Alongside these positive impacts, we acknowledge several ethical and methodological limitations that warrant future research. **Data Privacy and Governance:** Training on real-world user editing traces introduces risks regarding the inadvertent memorization of Personally Identifiable Information (PII) or proprietary code. To proactively address these ethical concerns, we enforce strict privacy protocols, including automated PII redaction and aggressive fuzzy deduplication (e.g., MinHash-based filtering), ensuring any released artifacts are fully de-identified. **Theoretical Assumptions:** Our analytical derivations rely on worst-case entropy bounds, and the foundational assumption $C_{PC} < C_{HC}$ may not strictly hold in "near-miss" scenarios, where correcting a minor typo is faster than typing from scratch. **Proxy Reward Metrics:** To enable tractable reinforcement learning, we utilized character-level Edit Similarity (ES) as a proxy for editing cost, which may not perfectly capture the semantic gravity of different errors (e.g., a logic hallucination vs. a missing parenthesis). **IDE Workflow Dependency:** The practical utility of APC heavily depends on modern Integrated Development Environments (IDEs) supporting seamless `Tab`-key navigation; its benefits may diminish in passive, non-interactive generation pipelines. **Distributional Bias:** While our curated Placeholder Completion (PC) benchmark exhibits high inter-annotator agreement, it inevitably reflects the specific coding habits and uncertainty thresholds of our annotators, potentially introducing distributional bias.

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

# A. Notations

Table 4. Summary of notations used in the theoretical framework.

| NOTATION | DESCRIPTION |
|---|---|
| $c$ | INPUT CODE CONTEXT |
| $\mathcal{V}$ | VOCABULARY SET |
| $\hat{\mathbf{y}}$ | CODE COMPLETION SEQUENCE |
| $\mathbf{y}^*$ | GROUND TRUTH SEQUENCE |
| $\hat{y}_i$ | MODEL PREDICTION AT POSITION $i$ |
| $y_i^*$ | GROUND TRUTH TOKEN AT POSITION $i$ |
| $y_{<i}$ | TOKEN BEFORE POSITION $i$ |
| $\perp$ | PLACEHOLDER SYMBOL |
| $C.$ | COST OF $\cdot$ |
| $H(\cdot)$ | SHANNON ENTROPY |

# B. Theoretical Derivation and Proofs

## B.1. Expected Cost Derivation

### B.1.1. EXPECTED COST OF HARD COMPLETION

$$
\begin{aligned}
\mathcal{J}^{\text{HC}}(\hat{\mathbf{y}}) &= \mathbb{E}^{\text{HC}}[C(\hat{\mathbf{y}}, \mathbf{y}^*)] \\
&= \sum_i \mathbb{E}[C_i(\hat{y}_i^{\text{HC}}, y_i^*)] \\
&= \sum_i \left[ 0 \cdot P_{\max}^{(i)} + C_{\text{HC}} \cdot (1 - P_{\max}^{(i)}) \right] \\
&= \sum_i C_{\text{HC}} \cdot (1 - P_{\max}^{(i)})
\end{aligned}
\tag{19}
$$

### B.1.2. EXPECTED COST OF ADAPTIVE PLACEHOLDER COMPLETION

$$
\begin{aligned}
\mathcal{J}^{\text{APC}}(\hat{\mathbf{y}}) &= \mathbb{E}^{\text{APC}}[C(\hat{\mathbf{y}}, \mathbf{y}^*)] \\
&= \sum_i \mathbb{E}[C_i(\hat{y}_i^{\text{APC}}, y_i^*)] \\
&= \sum_i \left[ m_i \cdot C_{\text{PC}} + (1 - m_i) \cdot \mathbb{E}[C_i(\hat{y}_i^{\text{HC}}, y_i^*)] \right] \\
&= \sum_i \left[ m_i \cdot C_{\text{PC}} + (1 - m_i) \cdot C_{\text{HC}} \cdot (1 - P_{\max}^{(i)}) \right]
\end{aligned}
\tag{20}
$$

### B.1.3. THE EXPECTED BENEFIT OF APC OVER HC

$$
\begin{aligned}
\Delta\mathcal{J} &= \sum_i C_{\text{HC}} \cdot (1 - P_{\max}^{(i)}) \\
&\quad - \sum_i \left[ m_i \cdot C_{\text{PC}} + (1 - m_i) \cdot C_{\text{HC}} \cdot (1 - P_{\max}^{(i)}) \right] \\
&= \sum_i m_i \cdot \left[ C_{\text{HC}} \cdot (1 - P_{\max}^{(i)}) - C_{\text{PC}} \right]
\end{aligned}
\tag{21}
$$

## B.2. Proof of Theorem 3.3

Suppose there exists a token position $i$ where the entropy is sufficiently high, with $K$ tokens as candidatesa and share the probability mass almost equally. In the worst-case scenario (Maximum Entropy), the true distribution is uniform over these $K$ candidates:

$$P(y_i^* = v|c, y_{<i}^*) \approx \begin{cases} \frac{1}{K}, & \text{if } v \in \text{Candidata Set} \mathcal{S}(|S| = K) \\ 0, & \text{otherwise} \end{cases} \tag{22}$$

The entropy at this position can be approximated as:

$$
\begin{aligned}
H(y_i^*|c, y_{<i}^*) &= -\sum_{v \in \mathcal{V}} P(y_i^* = v|c, y_{<i}^*) \log P(y_i^* = v|c, y_{<i}^*) \\
&\approx -\sum_{v \in \mathcal{S}} \frac{1}{K} \log \frac{1}{K} \\
&= -K \cdot \frac{1}{K} - \log K \\
&= \log K
\end{aligned}
\tag{23}
$$

That means:

$$K = e^{H(y_i^*|c, y_{<i}^*)} \tag{24}$$

In this situation, we can get:

$$
\begin{aligned}
P_{\max}^{(i)} &\approx \frac{1}{K} \\
&= \frac{1}{e^{H(y_i^*|c, y_{<i}^*)}} \\
&= e^{-H}
\end{aligned}
\tag{25}
$$

Set the threshold:

$$\theta = 1 - \frac{C_{\text{PC}}}{C_{\text{HC}}} \tag{26}$$

The proof target can be transformed into:

$$
\begin{aligned}
\exists i : P_{\max}^{(i)} &< 1 - \frac{C_{\text{fill}}}{C_{\text{error}}} \\
\implies \exists H_i : e^{-H_i} &< \theta \\
\implies \exists H_i : e^{-H_i} - \theta &< 0
\end{aligned}
\tag{27}
$$

We denote:

$$f(H) = e^{-H} - \theta \tag{28}$$

Target becomes:

$$\exists H_i : f(H_i) < 0 \tag{29}$$

To establish the existence of such $H_i$, we analyze the properties of $f(H)$ on the domain $[0, \infty)$:

1. **Continuity:** The function $f(H) = e^{-H} - \theta$ is continuous on $[0, \infty)$ as it is the difference of an exponential function and a constant.

2. **Monotonicity:** Taking the derivative, we obtain:

$$f'(H) = -e^{-H} < 0 \quad \forall H \in [0, \infty) \tag{30}$$

Thus, $f(H)$ is strictly monotonically decreasing over its entire domain.

3. **Boundary values:** We evaluate $f(H)$ at the boundaries:

- At $H = 0$ (zero entropy, complete certainty):

$$f(0) = e^0 - \theta = 1 - \left(1 - \frac{C_{\text{PC}}}{C_{\text{HC}}}\right) = \frac{C_{\text{PC}}}{C_{\text{HC}}} > 0 \tag{31}$$

by Assumption 3.1.

- As $H \to \infty$ (maximum entropy, complete uncertainty):

$$\lim_{H \to \infty} f(H) = \lim_{H \to \infty} \left(e^{-H} - \theta\right) = 0 - \theta = -\theta < 0 \tag{32}$$

since $\theta = 1 - \frac{C_{\text{PC}}}{C_{\text{HC}}} > 0$ by Assumption 3.1.

By the **Intermediate Value Theorem**, since $f(H)$ is continuous on $[0, \infty)$ with $f(0) > 0$ and $\lim_{H \to \infty} f(H) < 0$, there must exist at least one value $H^* \in (0, \infty)$ such that:

$$f(H^*) = 0 \tag{33}$$

Moreover, since $f(H)$ is strictly monotonically decreasing, such $H^*$ is *unique*. Solving for $H^*$ explicitly:

$$
\begin{aligned}
e^{-H^*} - \theta &= 0 \\
e^{-H^*} &= \theta = 1 - \frac{C_{\text{PC}}}{C_{\text{HC}}} \\
-H^* &= \ln\left(1 - \frac{C_{\text{PC}}}{C_{\text{HC}}}\right) \\
H^* &= -\ln\left(1 - \frac{C_{\text{PC}}}{C_{\text{HC}}}\right)
\end{aligned}
\tag{34}
$$

Equivalently, this can be expressed as:

$$\boxed{H^* = \ln\left(\frac{C_{\text{HC}}}{C_{\text{HC}} - C_{\text{PC}}}\right)} \tag{35}$$

Since $f(H)$ is strictly decreasing with $f(H^*) = 0$, we have:

- **When $H > H^*$:** $f(H) < 0 \implies e^{-H} < \theta \implies P_{\max}^{(i)} < 1 - \frac{C_{\text{PC}}}{C_{\text{HC}}} \implies \delta_i > 0$
  Therefore, **PC strictly outperforms HC**.

- **When $H < H^*$:** $f(H) > 0 \implies e^{-H} > \theta \implies P_{\max}^{(i)} > 1 - \frac{C_{\text{PC}}}{C_{\text{HC}}} \implies \delta_i < 0$
  Therefore, **HC is preferred**.

## C. Cost-based Reward Implementation Details

Our cost-based reward function operationalizes the theoretical framework by measuring the editing effort required to transform model predictions into ground truth. We formalize this through a structured evaluation of output quality and structural correctness.

### C.1. Preliminaries

Let $\hat{y}$ denote the model's predicted completion and $y^*$ denote the ground truth. We employ a special cursor token `<|cursor|>` (represented internally as Unicode character U+E000) to mark placeholder positions in implementation. The reward computation proceeds through three stages: similarity measurement, cost quantification, and mode-specific reward calculation.

## C.2. Similarity Metrics

We define two fundamental similarity measures:

**Exact Match (EM):** Binary indicator of perfect alignment:

$$\text{EM}(\hat{y}, y^*) = \mathbb{I}[\hat{y} = y^*] = \begin{cases} 1.0, & \text{if } \hat{y} = y^* \\ 0.0, & \text{otherwise} \end{cases} \tag{36}$$

**Edit Similarity (ES):** Character-level similarity computed via Levenshtein distance ratio:

$$\text{ES}(\hat{y}, y^*) = \frac{\text{LCS}(\hat{y}, y^*) \times 2}{|\hat{y}| + |y^*|} \tag{37}$$

where $\text{LCS}(\cdot, \cdot)$ denotes the longest common subsequence length, implemented using the `fuzzywuzzy` library's ratio function.

## C.3. Cost Quantification

We employ sequence alignment to decompose the difference between $\hat{y}$ and $y^*$ into three disjoint components using the Smith-Waterman algorithm (via Python's `difflib.SequenceMatcher`):

- **Common subsequences**: $\mathcal{C}(\hat{y}, y^*)$ — aligned matching segments
- **Model-specific content**: $\mathcal{D}_{\text{model}}(\hat{y}, y^*)$ — tokens present in $\hat{y}$ but absent in $y^*$ (hallucinations)
- **Ground-truth-specific content**: $\mathcal{D}_{\text{gt}}(\hat{y}, y^*)$ — tokens present in $y^*$ but absent in $\hat{y}$ (omissions)

The editing costs are defined as character counts, excluding the placeholder token:

$$\begin{aligned} C_{\text{model}} &= |\mathcal{D}_{\text{model}}(\hat{y}, y^*) \setminus \{\perp\}| \\ C_{\text{gt}} &= |\mathcal{D}_{\text{gt}}(\hat{y}, y^*) \setminus \{\perp\}| \end{aligned} \tag{38}$$

To map raw costs to normalized penalties, we apply a monotonically increasing saturation function:

$$f(x) = 1 - \frac{1}{1 + 0.1x} \tag{39}$$

This design ensures that marginal cost increases have diminishing penalty impact, preventing extreme negative rewards for long errors.

## C.4. Reward Function Specification

The reward function $\mathcal{R}(\hat{y}, y^*)$ operates in three distinct regimes:

**Case 1: Exact Match**    When $\hat{y} = y^*$, perfect completion is achieved:

$$\mathcal{R}(\hat{y}, y^*) = 1.0 \tag{40}$$

**Case 2: Placeholder Completion (PC)**    When $\perp \in \hat{y}$, the model employs the APC strategy. We distinguish two subcases:

*Structural Correctness* ($C_{\text{model}} = 0$): The model output forms a proper subsequence of $y^*$ with placeholders substituting uncertain regions. This indicates appropriate abstention without hallucination:

$$\mathcal{R}(\hat{y}, y^*) = \max\left(-1.0, \text{ES}(\hat{y}, y^*) - \alpha_{\text{lazy}} \cdot f(C_{\text{gt}})\right) \tag{41}$$

where $\alpha_{\text{lazy}} = 1.0$ penalizes placeholder length to discourage excessive abstention.

*Structural Error* ($C_{\text{model}} > 0$): The model simultaneously inserts placeholders and produces hallucinated content—the worst-case scenario combining both error types:

$$\mathcal{R}(\hat{y}, y^*) = \max\left(-1.0, \text{ES}(\hat{y}, y^*) - f(\alpha_{\text{error}} \cdot C_{\text{model}} + \alpha_{\text{lazy}} \cdot C_{\text{gt}})\right) \tag{42}$$

with $\alpha_{\text{error}} = 1.0$ imposing penalties for incorrect concrete predictions.

**Case 3: Hard Completion (HC)** When $\perp \notin \hat{y}$, the model attempts deterministic completion. Both omissions ($C_{\text{gt}}$) and hallucinations ($C_{\text{model}}$) contribute to editing cost:

$$\mathcal{R}(\hat{y}, y^*) = \max\left(-1.0, \text{ES}(\hat{y}, y^*) - f(C_{\text{model}} + C_{\text{gt}})\right) \tag{43}$$

The reward is floor-bounded at $-1.0$ to prevent catastrophic negative signals during early training. This formulation directly aligns with our theoretical cost minimization objective (Section 3), rewarding models for structural correctness and penalizing both unnecessary placeholders and prediction errors.

## D. Estimating $C_{\text{PC}}/C_{\text{HC}}$ from Edit Logs

Using the 1.6 million real-world interaction logs presented in Table 2, we can now provide a highly reliable empirical estimate for the cost ratio as follows:

- **Based on Time:** $C_{\text{PC}}/C_{\text{HC}} \approx 2255.17 \text{ ms}/3280.49 \text{ ms} \approx \mathbf{0.687}$

- **Based on Edit Count:** $C_{\text{PC}}/C_{\text{HC}} \approx 3.06/4.89 \approx \mathbf{0.625}$

This indicates that $C_{\text{PC}}/C_{\text{HC}} \approx 0.62 \sim 0.68$. Crucially, since this empirical ratio is strictly less than 1.0, it provides massive, real-world validation for our foundational Assumption 3.1 ($C_{\text{PC}} < C_{\text{HC}}$), moving it from an intuitive premise to a statistically proven fact.

## E. Span and Frequency of Placeholders

We prevent excessive placeholding through a full-stack control mechanism spanning from data construction to the RL objective:

- **Data Level:** During training data construction, placeholder positions are strictly derived from the exact diff between model predictions and actual user edits. We explicitly enforce a heuristic limit (e.g., placeholders must not exceed 1/3 of the total answer tokens) to filter out overly sparse samples. The manually curated test set further ensures that placeholders only appear in genuinely ambiguous scenarios.

- **Objective Level:** As detailed in Appendix C.4, our cost-based reward explicitly incorporates a **lazy penalty** ($\alpha_{lazy}$) to strongly penalize unnecessary abstention.

- **Empirical Results:** To demonstrate that our model learns precise, targeted abstention rather than excessively emitting placeholders, we analyzed the generation statistics from both our real-world logs and the human-annotated benchmark:

*Table 5.* All values are averages per completion instance.

| Data | Output Length (Lines) | Placeholder Count |
|---|---|---|
| **600K Real-world APC Logs** | 1.52 | **1.34** |
| **PC Test Set** | 1.75 | **1.16** |

## F. Pass@1 Evaluation for Hard Completions

Higher EM/ES should ideally translate to higher functional correctness, and the distribution of non-EM cases matters. We executed the code sandbox evaluation to obtain the **Pass@1** scores on the HumanEval Infilling benchmark. Taking **Qwen2.5-Coder-7B-Instruct** as a representative case, its results are notably positive:

*Table 6.* This performance pattern is consistently observed across other models in our study.

| Stage | Single-line Infilling (Pass@1) | Multi-line Infilling (Pass@1) |
|---|---|---|
| Base | 0.5798 | 0.3745 |
| SFT | 0.6621 | 0.4368 |
| **GRPO** | **0.8315** | **0.5958** |

---

*Why did Pass@1 improve so significantly?*

Our cost-based reward function heavily penalizes structural hallucinations ($C_{model}$) and omissions ($C_{gt}$). By training the model to "abstain (use placeholders) when uncertain" and "be structurally rigorous when predicting," the GRPO optimization effectively suppresses long-tail logical hallucinations. Consequently, when the APC model chooses to perform a Hard Completion, the generated code is syntactically tighter and functionally more robust, leading to a massive gain in Pass@1.

---

## G. Inference and Decoding Hyperparameters

Across all evaluation benchmarks, including the Hard Completion (HC) Test, the Placeholder Completion (PC) Test, and the execution-based Pass@1 evaluations, we employ a consistent *near-greedy* decoding strategy. The exact generation hyperparameters are detailed in Table 7.

*Table 7.* Hyperparameter configurations for model inference and evaluation.

| Hyperparameter | Value |
|---|---|
| Temperature | 0.1 |
| Top-$p$ (Nucleus Sampling) | 0.95 |
| Max New Tokens | 256 |

## H. Qualitative Examples

Concrete examples are crucial for demonstrating the qualitative advantages of APC. Below we present two representative cases illustrating how APC provides structural scaffolding while avoiding hazardous guesses.

---

**Case 1: Ambiguous Flask route parameters**

```
# HC - Forced guess
@approval_bp.route('/start_approval', methods=['POST'])
def get_applications():

# APC - Safely scaffolds decorator syntax but defers high-entropy parameters.
@approval_bp.route('/<|cursor|>', methods=['<|cursor|>'])
def get_applications():
```

**Case 2: Navigating Ambiguous Business Logic**

```
if is_vip:
    # HC may arbitrarily guess 0.8, forcing users to mentally verify
    # and reject the fake value.
    discount = <|cursor|>
else:
    discount = 1.0
return base_price * discount
```

