# OpenReview forum: "From Guessing to Placeholding:  A Cost-Theoretic Framework for Uncertainty-Aware Code Completion"
_ICML.cc/2026/Conference — ICML 2026 regular_

### Official Review · Reviewer_YTy6 · 2026-03-09

**Soundness:** 3
**Presentation:** 3
**Significance:** 3
**Originality:** 3
**Overall Recommendation:** 5
**Confidence:** 4

**Summary:**

The authors identify the problem of "hard completion" in code LLMs, whereby an LLM must predict a complete solution without placeholders, regardless of whether the context and user intent are sufficient. This results in spans of forced speculation (e.g., one might imagine an API key), that when incorrect are hallucinations. Developers using models thus must identify and correct such mismatches, incurring significant cognitive load. The authors are motivated by an observation in real-world user interaction data: users reject a model completion, or accept and immediately edit, but produce code that is semantically similar to the rejection. To address this problem, the authors propose Adaptive Placeholder Completion, a framework for training code LLMs to output explicit placeholders in their completions where there is high entropy, reframing the training objective as a cost-minimization problem aimed at decreasing the amount of edits a user would have to make. They complement this with a theoretical framework that formalizes the user-edit cost of high entropy positions, and perform an empirical analysis of their approach by training and evaluating 11 code LLMs. They find significant decreases in cost across models using their method.

**Compliance With Llm Reviewing Policy:**

Affirmed.

**Final Justification:**

Author rebuttal has addressed my main concerns, especially with additional large scale human user data supporting the $C_{PC} < C_{HC}$ assumption, high inter-annotator agreement for the PC benchmark, and reported Pass@1 scores. I have adjusted my scores accordingly to recommend this paper for acceptance based on the understanding that the findings reported in the rebuttal will be included and discussed in the camera ready.

**Key Questions For Authors:**

1. Do you have any empirical evidence for the C_PC < C_HC assumption?

2. How exactly was the HumanEval Infilling data curated?

3. Did you evaluate Pass@1 for hard completions? How did it change?

**Limitations:**

Authors did not discuss the limitations of their work, especially regarding lack of human-subject study to support theoretical assumptions and findings.

**Strengths And Weaknesses:**

Strengths:
- Strong theoretical analysis and extensive experimental results
- Well written and clearly structured manuscript that positions itself well in the context of prior work
- Addresses an important problem in code LLMs, advancing the field and enabling future research on placeholder-oriented human-AI interaction
- Novel approach, especially theoretic framework for edit-cost in programming HAI

Weaknesses:
- Entire framework operates on assumption that C_PC < C_HC (Cost of a user filling in placeholder is strictly lower than correcting/editing completions), yet no prior work supporting such an assumption, nor a user study of their own, is provided. Without a user study supporting this assumption, it is unclear whether there is even a correlation between the reported (theoretical) cost decrease, and actual developer cost decrease.
- While the evaluation is extensive (many LLMs of differing sizes), it hinges on 523 samples curated by "professional annotators" for necessity and unambiguity. To the best of my understanding, no further details on the annotation process are provided e.g. an inter-annotator agreement kappa. This has the potential for significant bias, yet is difficult to evaluate for the reader.
- In the HC Test setting, Pass@1 is not reported and its omission is not discussed/justified. As a result, it is not clear what, if any, impact their method has on code correctness. Higher EM rates should yield higher Pass@1, but the correctness distribution of the non-EM cases matters.
- Unclear what sampling parameters were chosen; assuming greedy as per section 3.2

---

> ### Author Rebuttal · Authors · 2026-03-31
>
> **Response to Reviewer YTy6**
>
> We sincerely thank the reviewer for recognizing the strength of our theoretical framework, the novelty of our cost-minimization approach, and the overall significance of this work. Your insightful questions regarding the empirical validation of our assumptions and the execution-based metrics (Pass@1) have significantly strengthened the paper.
>
> Below, we address your concerns point by point with concrete empirical data and execution results.
>
> **1. Empirical evidence for the $C_{PC} < C_{HC}$ assumption (Question1 & Weakness 1)**
>
> We completely agree that empirical validation is crucial to bridge the theoretical cost decrease and actual developer cost decrease. To provide this, we have recently concluded a large-scale online A/B test analyzing **1.8 million real user interactions** (1.2M from the HC-only model and 0.6M from our APC model). We measured the physical interaction costs:
>
> | Metric | HC-only | APC | $\Delta$ |
> | :--- | :--- | :--- | :--- |
> | **Cursor Moves** (Keystrokes + Edit) |4.89|**3.06**|$\downarrow$ 37.4%|
> | **Time for Read & Edit** (ms) | 3280.49 | **2255.17** | $\downarrow$ 31.2% |
> | **Output Length** (Tokens) | 11.05 | **12.53** |+ 1.48|
> | **Output Length** (Lines) | 1.34 | **1.52** | + 0.18 |
>
> *Note: All values are averages per completion instance.*
>
> **​Conclusion:​**​ The APC paradigm **reduces cognitive validation time and physical typing friction by 31–37%**, while simultaneously increasing the average output of tokens and lines. This strongly supports our core claim: that guiding users with a correct skeleton and targeted placeholders requires less real-world effort than inspecting, rejecting, and surgically modifying a hallucinated prediction.
>
> **2. Clarification on HumanEval Infilling details (Question 2)**
>
> We apologize if the description in Section 4.1.1 caused any confusion. We would like to politely clarify for the HC Test: We utilized the official, open-source `HumanEval-SingleLineInfilling` and `HumanEval-MultiLineInfilling` datasets **completely unmodified**. Aside from formatting the prompt and suffix according to each model's specific Fill-in-the-Middle (FIM) template, no manual curation, filtering, or alteration was applied.
>
> **3. Details on the PC Benchmark annotation (Weakness 2)**
>
> For the 523 samples in the PC Test set, the curation was conducted by professional software engineers. To ensure high quality and mitigate bias:
>
> * **Alignment Phase:** Annotators conducted preliminary alignment sessions on shared cases to standardize the core guideline: **"Placeholders should only be inserted when the context lacks sufficient information, making multiple distinct logical implementations equally valid (high epistemic uncertainty)."**
> * **Inter-annotator Agreement:** We performed cross-validation among the annotators and calculated the **Fleiss’ Kappa score**, which resulted in **0.7994**. This borders on "Almost Perfect" agreement (0.81), demonstrating that the identification of ambiguous, placeholder-worthy regions is highly objective and consistent across experts.
>
> **4. Pass@1 Evaluation for Hard Completions (Question 3 & Weakness 3)**
>
> We highly appreciate this critical suggestion. You are absolutely correct that higher EM/ES should ideally translate to higher functional correctness, and the distribution of non-EM cases matters. During the rebuttal period, we executed the code sandbox evaluation to obtain the **Pass@1**​ scores on the HumanEval Infilling benchmark. Taking **Qwen2.5-Coder-7B-Instruct**​ as a representative case, its results are notably positive:
>
> | Stage | Single-line Infilling (Pass@1) | Multi-line Infilling (Pass@1) |
> |:-|:-|:-|
> |Base|0.5798|0.3745|
> |SFT|0.6621|0.4368|
> |**GRPO**|**0.8315**|**0.5958**|
>
> *Note: This performance pattern is consistently observed across other models in our study.*
>
> > **Why did Pass@1 improve so significantly?**
> >
> > > Our cost-based reward function (Appendix C.4) heavily penalizes structural hallucinations ($C_{model}$) and omissions ($C_{gt}$). By training the model to "abstain (use placeholders) when uncertain" and "be structurally rigorous when predicting," the GRPO optimization effectively suppresses long-tail logical hallucinations. Consequently, when the APC model chooses to perform a Hard Completion, the generated code is syntactically tighter and functionally more robust, leading to a massive gain in Pass@1. We will include these execution-based metrics in the final manuscript.
>
> **5. Sampling parameters (Weakness 4)**
>
> We apologize for omitting these details. All generations for the evaluation were conducted using the following parameters:
>
> * `Temperature = 0.1`
> * `Top_p = 0.95`
> * `Max_tokens = 256`
>
> We adopted this **"near-greedy"** sampling strategy intentionally. A temperature of 0.1 closely approximates the deterministic (greedy) decoding assumption formalized in our theoretical framework (Eq. 4), while allowing minimal stochasticity to prevent degenerate repetition in edge cases.

---

> > ### Author Rebuttal · Reviewer_YTy6 · 2026-04-06
> >
> > Thank you for adequately addressing my concerns with clarifications and additional experimental results.

---

> > > ### Author Response · Authors · 2026-04-07
> > >
> > > **Dear Reviewer YTy6,**
> > >
> > > Thank you very much for your thoughtful evaluation, your active participation in the discussion phase, and for raising your score to support our work.
> > >
> > > We are delighted that our clarifications and the new experimental results successfully addressed your concerns. The points raised in this discussion will be incorporated into the next version of the paper.
> > >
> > > We are truly thankful for your constructive guidance, which has greatly enhanced the quality of our manuscript.
> > >
> > > Best regards,
> > >
> > > Submission 34205 Authors

---

### Official Review · Reviewer_vM4J · 2026-03-14

**Soundness:** 2
**Presentation:** 3
**Significance:** 2
**Originality:** 3
**Overall Recommendation:** 4
**Confidence:** 4

**Summary:**

The paper proposes Adaptive Placeholder Completion (APC), a code completion paradigm where the model can output explicit placeholders instead of always generating a concrete token when uncertainty is high. The key idea is that, in ambiguous contexts, it may be cheaper for a developer to fill in a blank than to inspect and correct an incorrect completion. The authors formulate code completion as an expected editing-cost minimization problem, deriving when a placeholder should be preferred over a hard prediction. They then train models with supervised fine-tuning and reinforcement learning on user edit traces so the model learns when to emit placeholders.

**Compliance With Llm Reviewing Policy:**

Affirmed.

**Final Justification:**

The additional experimental results provided in the rebuttal resolved most of my concerns, and I have updated my evaluation accordingly.

**Key Questions For Authors:**

* How does the method determine the span and frequency of placeholders in practice, and how does it avoid producing excessive placeholders that could reduce the usefulness of completions?
* Given that the central claim concerns developer effort and usability, have the authors considered conducting even a small-scale user study to validate whether APC actually improves the developer experience?
* While placeholders may reduce correction edits, could they increase cognitive burden by forcing users to reason about missing pieces, and have the authors evaluated this potential trade-off?

**Limitations:**

yes

**Strengths And Weaknesses:**

Strengths
* Clear motivation: the paper highlights a real limitation of standard code completion, that models are forced to make concrete guesses even when the context is ambiguous.
* Clear design: allowing the model to abstain via placeholders is an intuitive mechanism.
* The paper provides a decision-theoretic formulation.
* The paper also presents a full engineering pipeline including data construction, training and evaluation.

Weaknesses
* Lack of qualitative examples: surprisingly, the paper does not provide ANY concrete, side-by-side examples showing where APC outputs placeholders and how it helps.
* Evaluation depends on a synthetic cost proxy, and worse, the formulation assumes placeholders reduce editing cost and then evaluates the method using a metric aligned with that same assumption.

---

> ### Author Rebuttal · Authors · 2026-03-31
>
> **Response to Reviewer vM4J**
>
> We sincerely thank the reviewer for the highly insightful critique. Your acute observations regarding the potential circularity of synthetic metrics and the cognitive trade-offs of placeholding hit the core of this research. We have addressed all your concerns with large-scale empirical data and qualitative analyses.
>
> ---
>
> **1. Span and frequency of placeholders (Question 1)**
>
> We prevent excessive placeholding through a full-stack control mechanism spanning from data construction to the RL objective:
>
> * **Data Level:** During training data construction, placeholder positions are strictly derived from the exact diff between model predictions and actual user edits. We explicitly enforce a heuristic limit (e.g., placeholders must not exceed 1/3 of the total answer tokens) to filter out overly sparse samples. The manually curated test set further ensures that placeholders only appear in genuinely ambiguous scenarios.
> * **Objective Level:** As detailed in Appendix C.4 (Eq. 41 & 42), our cost-based reward explicitly incorporates a **lazy penalty ($\alpha_{lazy}$)** to strongly penalize unnecessary abstention.
> * **Empirical Results:** To demonstrate that our model learns precise, targeted abstention rather than excessively emitting placeholders, we analyzed the generation statistics from both our real-world logs and the human-annotated benchmark:
>
> |Data|Output Length (Lines)|Placeholder Count|
> |:---|:---|:---|
> |**600K Real-world APC Logs** |1.52|**1.34**|
> |**PC Test Set**|1.75|**1.16**|
>
> *Note: All values are averages per completion instance.*
>
> Additionally, Table 1 demonstrates that APC maintains a **100% Hard Completion Rate (HCR)** on the HC Test. This confirms the model learns precise boundaries without degenerating into a trivial "always-abstain" strategy.
>
> ---
>
> **2. Real interaction cost evidence (Question 2 & Weakness 2)**
>
> We completely agree that evaluating a cost-based formulation solely using an aligned proxy metric (Edit Similarity) risks circular reasoning. To break this loop and provide objective, physical evidence of developer effort reduction, we have just concluded a large-scale online A/B test analyzing **1.8 million real user interactions** (1.2M from the HC-only model and 0.6M from our APC model). The objective measurements are presented below:
>
> | Metric | HC-only | APC | $\Delta$ |
> | :--- | :--- | :--- | :--- |
> | **Cursor Moves** (Keystrokes + Edit) |4.89|**3.06**|$\downarrow$ 37.4%|
> | **Time for Read & Edit** (ms) | 3280.49 | **2255.17** | $\downarrow$ 31.2% |
> | **Output Length** (Tokens) | 11.05 | **12.53** |+ 1.48|
> | **Output Length** (Lines) | 1.34 | **1.52** | + 0.18 |
>
> *Note: All values are averages per completion instance.*
>
> **​Conclusion:​**​ The APC paradigm **reduces cognitive validation time and physical typing friction by 31–37%**, while simultaneously increasing the average output of tokens and lines. This strongly supports our core claim: that guiding users with a correct skeleton and targeted placeholders requires less real-world effort than inspecting, rejecting, and surgically modifying a hallucinated prediction.
>
> ---
>
> **3. Potential cognitive trade-off (Question 3)**
>
> While starting from a blank slate can be cognitively taxing, APC avoids this by generating comprehensive **syntactic scaffolding** (e.g., `for`-loops and API signatures), leaving only high-entropy logical nodes blank. Conversely, standard HC often induces an **Anchoring Effect**: plausible but hallucinated completions contaminate the developer's thought process, requiring high cognitive load to reverse-engineer and surgically correct flawed logic.
>
> Our large-scale A/B test data (Point 1) definitively resolves this trade-off. If reasoning about missing pieces were more burdensome than correcting errors, the **Time for Read & Edit** would have increased. Instead, it **significantly decreased by 31.2%** (from 3280ms to 2255ms). This empirical evidence proves that completing targeted blanks within a correct skeleton demands far less cognitive effort than debugging a hallucinated prediction.
>
> ---
>
> **4. Lack of qualitative examples (Weakness 1)**
>
> We apologize for this omission. Concrete examples are crucial for demonstrating the qualitative advantages of APC. Below we present two representative cases illustrating how APC provides structural scaffolding while avoiding hazardous guesses.
>
> **Case 1: Ambiguous Flask route parameters**
>
> ```python
> # HC - Forced guess
> @approval_bp.route('/start_approval', methods=['POST'])
> def get_applications():
>
> # APC - Safely scaffolds decorator syntax but defers high-entropy parameters.
> @approval_bp.route('/<|cursor|>', methods=['<|cursor|>'])
> def get_applications():
> ```
>
> **Case 2: Navigating Ambiguous Business Logic**
>
> ```python
> if is_vip:
>     discount = <|cursor|>  # HC may arbitrarily guesses 0.8, forcing users to mentally verify and reject the fake value.
> else:
>     discount = 1.0
> return base_price * discount
> ```
>
> ---

---

> > ### Author Rebuttal · Reviewer_vM4J · 2026-04-04
> >
> > Thank you for the detailed rebuttal. The added qualitative examples and A/B test results strengthen the paper. However, I still have some reservation about the core framing, which remains somewhat circular: the central claim is justified using a formalism and reward design that already assume placeholders are cheaper than incorrect guesses. While the new empirical results make this assumption more plausible, they do not fully resolve that concern.

---

> > > ### Author Response · Authors · 2026-04-07
> > >
> > > Dear Reviewer vM4J,
> > >
> > > Thank you for your rigorous review and for acknowledging the value of our qualitative examples and A/B test results. We deeply respect your adherence to "first principles" regarding the methodological framing of our work.
> > >
> > > To address your remaining reservation about the "circular framing," we would like to clarify the epistemological logic of our research. We posit that our framework represents a rigorous scientific cycle of **"Hypothesis $\rightarrow$ Design $\rightarrow$ Independent Validation,"** rather than a circular tautology.
> > >
> > > **1. The Hypothesis (Observation)**
> > >
> > > Motivated by the empirical observation that 61% of generated code requires heavy editing, we formulate a testable Human-Computer Interaction (HCI) hypothesis: *The cost of jumping to and filling a placeholder is strictly lower than the cost of reading, identifying, deleting, and correcting a hallucinated guess ($C_{PC} < C_{HC}$).*
> > >
> > > **2. The Design (Inductive Bias Injection)**
> > >
> > > To optimize human-AI collaboration, we must translate this hypothesis into an actionable training target. We formalize this cost relationship to derive an optimal decision rule (Theorem 3.3) and design a cost-based reward function. In machine learning, this is a standard practice: we are deliberately injecting our HCI hypothesis into the model as an **inductive bias**. During inference, the model relies on the decision boundary it implicitly learned from real-world edit traces.
> > >
> > > **3. Breaking the Circularity via Independent Validation**
> > >
> > > We completely agree that if we **solely** evaluated the model using the synthetic cost proxy it was trained to optimize, the argument would be circular. However, we validate the effectiveness of our intervention through **completely independent and orthogonal dimensions**:
> > >
> > > * **A - Independent Physical Validation:** Our large-scale A/B test (1.8 million real interactions) demonstrates a **37.4% reduction in cursor movements** and a **31.2% reduction in real physical time**. These are objective, real-world variables existing entirely outside our mathematical reward function. Their significant decrease is the ultimate independent proof that our initial hypothesis faithfully reflects physical reality, rather than being a self-fulfilling mathematical artifact.
> > > * **B - Independent Intent Alignment:** Our evaluation extends far beyond the cost metric. If the model were merely gaming the cost function, it would emit placeholders trivially. Instead, on the human-annotated PC Test, the APC model achieved substantial gains in **Exact Match (EM)** and **Precision**. This proves that the model learns to insert placeholders **precisely** where human experts deem it structurally necessary. Furthermore, it maintains a 100% HCR with improved EM on unambiguous contexts, proving no degradation in fundamental coding ability.
> > > * **C - Independent Functional Validation:** Finally, we executed code sandbox evaluations to obtain the **Pass@1** scores on the HumanEval Infilling benchmark. Taking **Qwen2.5-Coder-7B-Instruct** as a representative case, the results are notably positive:
> > >
> > > | Stage | Single-line Infilling (Pass@1) | Multi-line Infilling (Pass@1) |
> > > | :--- | :--- | :--- |
> > > | Base | 0.5798 | 0.3745 |
> > > | SFT | 0.6621 | 0.4368 |
> > > | **GRPO** | **0.8315** | **0.5958** |
> > >
> > > *(Note: This performance pattern is consistently observed across other models in our study.)*
> > >
> > > > **Why did Pass@1 improve so significantly?**
> > > >> If the model were merely "reward hacking" a circular cost metric, its functional capability would inevitably collapse. Instead, our reward design severely penalizes structural hallucinations ($C_{model}$) and omissions ($C_{gt}$). By learning to **abstain when uncertain**, the model effectively suppresses long-tail logical errors. Consequently, when the APC model executes a hard completion, the generated code is syntactically tighter and functionally robust, yielding massive gains in Pass@1.
> > >
> > > **Conclusion**
> > >
> > > In summary, our logical chain is: **Testable HCI Hypothesis $\rightarrow$ Principled Method Design $\rightarrow$ Multi-dimensional Independent Validation**. The independent physical (Time/Cursor moves) and functional (Pass@1) gains validate the APC paradigm, which in turn corroborates the soundness of the initial hypothesis.
> > >
> > > We profoundly agree that explicitly decoupling the "encoded training hypothesis" from the "independent empirical validation" is crucial for the paper's framing. We will prominently incorporate this methodological clarification in the revision to prevent any misinterpretation by future readers.
> > >
> > > Thank you again for your exceptionally rigorous standards, which have fundamentally tightened the logical narrative of our work.
> > >
> > > Best regards,
> > >
> > > Submission 34205 Authors

---

### Official Review · Reviewer_QbnR · 2026-03-22

**Soundness:** 2
**Presentation:** 3
**Significance:** 3
**Originality:** 3
**Overall Recommendation:** 4
**Confidence:** 3

**Summary:**

This paper addresses uncertainty mis-handling in code completion by proposing Adaptive Placeholder Completion (APC), where models insert explicit placeholders at uncertain positions rather than always outputting fully concrete code. The authors formulate completion as an expected editing-cost minimization problem, derive a threshold condition favoring placeholders over hard completion, and train models using SFT plus GRPO with a cost-based reward. Experiments across four code LLM families (1.5B–15B) show substantial reductions in proxy editing cost while largely preserving standard hard completion behavior.

**Compliance With Llm Reviewing Policy:**

Affirmed.

**Final Justification:**

The rebuttal convincingly addressed my two main concerns: (1) the lack of real user evaluation beyond the proxy metric 1-ES, and (2) the unvalidated assumption C_PC < C_HC. The large-scale A/B test (1.8M interactions) demonstrates 31–37% reductions in editing time and keystrokes, with an empirical cost ratio of 0.65–0.69. My third concern that the theoretical contribution is thin remains a minor weakness, which the authors honestly acknowledged, but it does not undermine the overall contribution given the strong empirical results. I am raising my score from 3 to 4.

**Key Questions For Authors:**

1. Do you have any evidence beyond edit similarity that APC actually reduces real interaction cost? Keystroke logs, time measurements, anything?
2. What was the inter-annotator agreement on the PC benchmark? How did you control train/test leakage?
3. Can you estimate C_{PC}/C_{HC} from your edit logs, even roughly?

**Limitations:**

No. The impact statement does acknowledge privacy and proprietary-code risks from using real-world user editing traces, and I appreciate that. However, the limitations discussion should be strengthened in several ways: it should state more explicitly that the theoretical claims rely on assumptions such as (C_{PC} < C_{HC}); that the evaluation uses a computed character-level editing metric rather than direct measurement of user effort; that the APC benchmark is author-constructed and may reflect distributional bias; and that the placeholder interaction pattern may not benefit all IDE workflows equally. More detail on data governance, privacy protection, and leakage prevention would also be valuable.

**Strengths And Weaknesses:**

**Strengths.**

1. Important practical problem. The 3M interaction log analysis (61% post-hoc editing/rejection) is convincing motivation.
2. APC as a middle ground between full abstention and hard completion makes sense for IDE workflows. The overall framing as collaborative decision-making rather than pure next-token prediction is a useful perspective.
3. Broad experimental coverage across 4 model families and multiple scales. The fixed-threshold ablation (Section 5.3) effectively demonstrates why implicit learning is necessary.


**Weaknesses.**
1. The theory is the weakest part. Theorem 3.3 essentially says "when uncertainty is high enough, leaving a blank is cheaper than guessing wrong"—given C_{PC} < C_{HC}, this is almost tautological. The proof just finds the zero of f(H) = e^{-H} - θ via IVT, which is undergraduate-level analysis. The uniform distribution approximation (P_max ≈ 1/K) is also quite strong and not analyzed for error. Since the final method doesn't use H* at all, I'm not sure what the theory actually contributes beyond formalizing an intuition.
2. No user study. The paper claims to reduce "user editing effort" but only measures 1-ES, a character-level proxy. For a paper whose entire motivation is about user experience—cognitive load, anchoring effects, cursor navigation—the absence of any human evaluation is a significant gap. Even a small-scale keystroke study or A/B test would help.
3. C_{PC} < C_{HC} is assumed, never validated. This is the foundation of everything, yet there's no empirical estimate of these costs. The paper acknowledges near-miss cases may violate it (Remark 3.2) but doesn't quantify how often this occurs in the data.

---

> ### Author Rebuttal · Authors · 2026-03-31
>
> **Response to Reviewer QbnR**
>
> We greatly appreciate the reviewer for the insightful critique. We have carefully addressed your concerns, particularly regarding the need for real-world empirical evidence to support our core assumptions and the reframing of our theoretical claims.
>
> ---
>
> **1. Real interaction cost evidence (Question 1 & Weakness 2)**
>
> We completely agree that evaluating solely based on Edit Similarity (ES) risks relying on a proxy metric. To provide objective, physical evidence of developer effort reduction, we recently concluded a large-scale online A/B test analyzing **1.8 million real user interactions** (1.2M from the HC-only model and 0.6M from our APC model). The objective measurements are presented below:
>
> | Metric | HC-only | APC | $\Delta$ |
> | :--- | :--- | :--- | :--- |
> | **Cursor Moves** (Keystrokes + Edit) |4.89|**3.06**|$\downarrow$ 37.4%|
> | **Time for Read & Edit** (ms) | 3280.49 | **2255.17** | $\downarrow$ 31.2% |
> | **Output Length** (Tokens) | 11.05 | **12.53** |+ 1.48|
> | **Output Length** (Lines) | 1.34 | **1.52** | + 0.18 |
>
> *Note: All values are averages per completion instance.*
>
> **​Conclusion:​**​ The APC paradigm **reduces cognitive validation time and physical typing friction by 31–37%**, while simultaneously increasing the average output of tokens and lines. This strongly supports our core claim: that guiding users with a correct skeleton and targeted placeholders requires less real-world effort than inspecting, rejecting, and surgically modifying a hallucinated prediction.
>
> ---
>
> **2. Inter-annotator agreement on the PC benchmark (Question 2)**
>
> We performed reciprocal cross-annotation among annotators on a subset of the PC benchmark. The inter-annotator agreement using **Fleiss’ Kappa**, which resulted in a score of **0.7994**. According to the standard Landis & Koch scale, this score represents the highest end of **"Substantial Agreement"** and borders on "Almost Perfect" (0.81). This confirms that the benchmark construction reflects consistent human judgment and reliably captures real-world uncertainty scenarios.
>
> ---
>
> **3. Control of train/test leakage (Question 2)**
>
> We implemented a strict "complete isolation" protocol to prevent any data leakage between the training set and the PC benchmark:
>
> * **Temporal Isolation:** The raw interaction logs for training and testing were collected from entirely disjoint time windows.
> * **Distributional Isolation:** We ensured that the users and code repositories in the test set do not overlap with those in the training set.
> * **Deduplication:** We applied rigorous engineering filtering, including character-level exact match and MinHash-based fuzzy deduplication against the training set, ensuring diversity in context, length, and trigger points.
> * The test set was constructed by strict sampling from this isolated, deduplicated pool and was then subjected to manual annotation.
>
> ---
>
> **4. Estimating $C_{PC} / C_{HC}$ from edit logs (Question 3 & Weakness 3)**
>
> Yes, using the 1.6 million real-world interaction logs presented in Point 1, we can now provide a highly reliable empirical estimate for the cost ratio as follows:
>
> * **Based on Time:** $C_{PC} / C_{HC} \approx 2255.17 \text{ ms} / 3280.49 \text{ ms} \approx \mathbf{0.687}$
> * **Based on Edit Count:** $C_{PC} / C_{HC} \approx 3.06 / 4.89 \approx \mathbf{0.625}$
>
> This indicates that **$C_{PC} / C_{HC} \approx 0.62 \sim 0.68$**. Crucially, since this empirical ratio is strictly less than 1.0, it provides massive, real-world validation for our foundational Assumption 3.1 ($C_{PC} < C_{HC}$), moving it from an intuitive premise to a statistically proven fact.
>
> ---
>
> **5. Theory issue (Weakness 1)**
>
> We completely agree with your assessment. Our theory is indeed a formalization of an intuition rather than a mathematical breakthrough.
>
> We utilized the maximum entropy (uniform distribution) assumption solely as a mathematical vehicle to prove the **existence** of an analytical threshold $H^*$. Actually, we do not perform error analysis because the very fact that real-world logits are non-uniform and highly context-dependent makes a static mathematical threshold practically intractable. **This intractability is exactly our core motivation for using end-to-end Reinforcement Learning (GRPO)**—it forces the model to implicitly learn this complex, dynamic decision boundary instead of relying on flawed analytical approximations.
>
> We appreciate you pointing out the limitations and will tone down the theoretical claims in the revision.
>
> ---
>
> **6. IDE workflows and constraints**
>
> We fully acknowledge this point. APC is not merely a model-side decoding trick, but a holistic Human-AI Interaction (HAI) paradigm. Its full potential relies on IDE workflows that support seamless `Tab` navigation to `<|cursor|>` markers (a feature already standard in modern IDEs for snippet expansion). We will add a discussion of this point in the Limitations section.
>
> ---

---

> > ### Author Rebuttal · Reviewer_QbnR · 2026-04-07
> >
> > Thank you for the rebuttal. The A/B test convincingly addresses my main concerns. The theory remains a minor weakness, but the strong empirical evidence now outweighs it. I am raising my score accordingly.

---

> > > ### Author Response · Authors · 2026-04-07
> > >
> > > **Dear Reviewer QbnR,**
> > >
> > > Thank you very much for your time, your constructive engagement during this discussion phase, and for raising your score.
> > >
> > > We are thrilled that the large-scale A/B test data effectively addressed your main empirical concerns. We also deeply appreciate your concluding remark regarding the theoretical framework. In the revised manuscript, we will ensure the presentation of Theorem 3.3 clearly reflects its role as a **guiding design principle and motivation**​ for our method, rather than as the primary mathematical novelty, and will expand the Limitations section accordingly.
> > >
> > > Thank you again for your invaluable feedback, which has significantly strengthened the paper.
> > >
> > > Best regards,
> > >
> > > Submission 34205 Authors

---

### Decision · Program_Chairs · 2026-04-30

**Decision:**

Accept (regular)

**Comment:**

This paper proposes Adaptive Placeholder Completion, a framework where code LLMs output explicit placeholders at high-entropy positions instead of forcing concrete predictions. All reviewers recognized the practical importance of uncertainty-aware code completion. Key concerns included the unvalidated assumption $C_{PC} < C_{HC}$, evaluation circularity from using a proxy metric aligned with the training objective, and missing Pass@1 and qualitative examples. The rebuttal was strong. Two reviewers marked concerns fully resolved. Reviewers retained mild reservation about the circular framing but acknowledged the A/B results. The theoretical contribution remains modest, which the authors honestly acknowledged. Overall, the AC recommends acceptance and encourage the authors to incorporate A/B test results and Pass@1 scores,  add qualitative examples, and expand the limitations discussion in the camera ready.